

# Filipino sign language alphabet recognition using Persistent Homology Classification algorithm

Cristian B. Jetomo and Mark Lexter D. De Lara

Institute of Mathematical Sciences, College of Arts and Sciences, University of the Philippines Los Baños, Los Baños, Laguna, Philippines

## ABSTRACT

Increasing number of deaf or hard-of-hearing individuals is a crucial problem since communication among and within the deaf population proves to be a challenge. Despite sign languages developing in various countries, there is still lack of formal implementation of programs supporting its needs, especially for the Filipino sign language (FSL). Recently, studies on FSL recognition explored deep networks. Current findings are promising but drawbacks on using deep networks still prevail. This includes low transparency, interpretability, need for big data, and high computational requirements. Hence, this article explores topological data analysis (TDA), an emerging field of study that harnesses techniques from computational topology, for this task. Specifically, we evaluate a TDA-inspired classifier called Persistent Homology Classification algorithm (PHCA) to classify static alphabet signed using FSL and compare its result with classical classifiers. Experiment is implemented on balanced and imbalanced datasets with multiple trials, and hyperparameters are tuned for a comprehensive comparison. Results show that PHCA and support vector machine (SVM) performed better than the other classifiers, having mean Accuracy of 99.45% and 99.31%, respectively. Further analysis shows that PHCA's performance is not significantly different from SVM, indicating that PHCA performed at par with the best performing classifier. Misclassification analysis shows that PHCA struggles to classify signs with similar gestures, common to FSL recognition. Regardless, outcomes provide evidence on the robustness and stability of PHCA against perturbations to data and noise. It can be concluded that PHCA can serve as an alternative for FSL recognition, offering opportunities for further research.

# INTRODUCTION

One of the top causes of disability in the global scale is hearing loss. There are 466 million people estimated to have this disability according to the World Health Organization (*Davis & Hoffman, 2019*). It is projected that by 2050, nearly 2.5 billion people will have some degree of hearing loss and at least 700 million people will require hearing rehabilitation.

Corresponding author
Cristian B. Jetomo,
cbjetomo@up.edu.ph

Communication is an essential component to human existence, but is largely inaccessible to the deaf community. In turn, hard-of-hearing individuals struggle in a lot of crucial aspects such as education (*Most, 2004*), employment (*Cruz & Calimpusan, 2018*), access to healthcare (*Souza et al., 2017*), limited social interactions especially during the COVID-19 pandemic (*Singh et al., 2021*), and more.

To cope with this, sign languages have emerged across different countries. These languages are performed through manual signals (*e.g.*, hand gestures, dynamic movements) and non-manual signals (*e.g.*, facial expressions, body motions, and palm orientations) to properly portray the corresponding meaning of the signed gesture (*Rivera & Ong, 2018*). Sign languages have been effective for this purpose, and hence, have been the primary means of communication for the deaf community. In the Philippine context, Filipino sign language (FSL) has been declared as its national sign language and is mandated to be recognized, promoted, and supported in all transactions. Numerous public and private programs are being conducted to increase the number of individuals that are proficient in FSL, and in general, sign language interpretation. However, there is still a lack of formal implementation of the programs, furthering the gap between the normal hearing and hard-of-hearing population of the country. With this, a better solution is needed.

Currently, researchers leveraged the use of machine learning to automate the interpretation process of signed gestures. *Adeyanju, Bello & Adegboye (2021)* provided a comprehensive review of methods applied to sign language recognition and an overview of feature extraction and preprocessing techniques. The authors highlighted that the trend in publication of sign language recognition articles is consistent with the increase in people having hearing disability. In *Jain et al. (2021)*, support vector machines and convolutional neural networks were utilized to recognize the American sign language alphabet. *Samaan et al. (2022)* used three recurrent neural networks to recognize 10 dynamic sign languages where features are extracted using MediaPipe. In *Buttar et al. (2023)*, two deep networks, YOLOv6 and long-short term memory, were explored to develop a hybrid algorithm that can classify both static and dynamic signs. *Martínez-Hinarejos & Parcheta (2017)* used continuous density hidden Markov model to recognize basic sentences using the Spanish sign language instead of isolated words or phrases. These articles give insight as to how vast areas of mathematics (*e.g.*, statistics, linear algebra, and calculus) are being explored to solve sign language problems. This raises a question whether other fields such as computational topology can also contribute to solve the problem, specifically for FSL. We attempt to answer this question in this article.

Topological data analysis (TDA) is a newly emerging field that harnesses techniques from computational topology to analyze data. It makes use of these concepts to extract shape or topological features, usually *via* Persistent Homology, which is important for machine learning problems. TDA is favored by researchers since it is easy to scale for larger datasets due to the stability of Persistent Homology under perturbations and noise (*Hensel, Moor & Rieck, 2021*; *Mishra & Motta, 2023*). The field is beginning to get attention for different applications such as rhythm detection (*Ness-Cohn & Braun, 2021*), biomedicine (*Skaf & Laubenbacher, 2022*), financial market analysis (*Basu & Li, 2019*), aviation (*Li, Ryerson & Balakrishnan, 2019*), and more. It is also starting to be explored for sign

language recognition. In *Mirehi, Tahmasbi & Targhi (2019)*, graph structures are constructed from the hand and meaningful shape features are extracted from topological properties of the graph. This makes the features stable against different deformations, scale, and noise. *Özdemir, Baytas & Akarun (2024)* investigates the optimal choices for hand graph topology by adopting the spatio-temporal graph convolutional networks. However, it can be noted that articles that directly implement TDA for this problem remains limited.

TDA can be applied for machine learning problems in many approaches. A survey is conducted by *Hensel, Moor & Rieck (2021)* to review and synthesize the current state of the fuse of the two areas. The authors divided some of these applications into two parts: extrinsic and intrinsic approach. Extrinsic topological approach utilizes persistent homology to obtain a representation of data in the form of persistence diagrams. These diagrams are converted into features, using either vector-based or kernel-based representations, which are then fed into standard machine learning models. On the other hand, intrinsic topological approach incorporates TDA in the machine learning model itself. This approach either includes topological information into the design of the model or harness topological methods to study and improve the model. One example of the latter approach is the novel classifier called Persistent Homology Classification algorithm (PHCA) (*De Lara, 2023*).

PHCA has been applied on different variants of datasets and have shown to perform at par if not better than the majority of classical classifiers including support vector machines (SVM), random forest (RF), K-nearest neighbors (KNN), linear discriminant analysis (LDA), and classification and regression trees (CART) (*De Lara, 2023*). The Iris Plant, Wheat Seed, Social network ads, MNIST handwritten digits, and synthetic datasets are used to evaluate PHCA in comparison with these classical classifiers. Results show that the performance of PHCA is not significantly different from that of the other classifiers, even exceeding a few of them in some datasets.

Now, this study aims to extend the capability of the computational topology-based classifier for FSL recognition. Specifically, it aims to utilize PHCA to classify images of hands signing the FSL alphabet and compare its result with classical classifiers. Additionally, PHCA and the classical classifiers are tested on balanced and imbalanced datasets. Results show that PHCA is one of the best performing classifier for this task, obtaining a mean accuracy of up to 99.31%. This provides insights to the potential of TDA as an alternative method for sign language recognition.

## RELATED WORKS

In the previous section, we introduced some articles on sign language recognition. Now, we discuss the current state of FSL recognition and highlight the development of solutions for the problem.

One of the earliest accounted articles that dealt with FSL recognition is by *Sandjaja & Marcos (2009)*. The authors used color-coded gloves to efficiently extract the position of the fingers using a multi-color tracking algorithm. From this, features are extracted and the data is recognized using Hidden Markov Model. Of the 5,000 FSL number videos, they

were able to obtain a highest accuracy of 85.52%. This is an excellent first step for FSL recognition but is inefficient since a color-coded glove is needed. Following this, *Cabalfin et al. (2012)* utilized a manifold projection approach in classifying 72 common FSL signs. From the training set, reference manifolds are created using the Isomap algorithm. Then, signs are transformed into trajectories using these manifolds. These trajectories are then compared by selecting the closest reference trajectory using dynamic time warping and longest common subsequence. The authors achieved 89% as their highest accuracy. The authors' contribution is on the exploration of manifold learning for recognizing FSL. However, results show that their method has trouble differentiating signs with similar movements and is inefficient for a general recognition system.

A different approach is implemented by *Oliva et al. (2018)* to classify 10 basic Filipino words. The authors used a Kinect sensor which captures the location, movement, and audio of a person. Some joints of interests are considered in the article to eliminate redundancy in the features. The features are projected onto the Cartesian and Spherical coordinate system and classified using dynamic time warping and support vector machines. The article obtained a peak accuracy of 95%, recall of 95%, and precision of 95.89%. The authors highlight that the use of Spherical coordinates showed consistency in performance regardless of size and location of the signer and have also noted tendency to misclassify similar signs. *Rivera & Ong (2018)* used the same sensor to extract data but explored the importance of non-manual signals for FSL recognition. They focus on face orientations, shape units, and animation units as features and captured the movement of the eyes, eyebrows, mouth, nose, and head. Classification is performed using support vector machine and artificial neural networks, obtaining the highest accuracy of 87% on the emotion class. These articles are promising but is non-conventional since a specific sensor is required for recognition.

As machine learning progressed and data became easily available, so are the methods for FSL recognition. Deep networks are now being used for the task. In *Montefalcon, Padilla & Llabanes Rodriguez (2021)*, two residual networks, ResNet-18 and ResNet-50, are explored to classify 10,000 image-formatted FSL numbers. The images were preprocessed with and without Gaussian Blur. Outcomes show that using this technique improves the result. They concluded that ResNet-18 leads to over-fitting, having excellent results on the training but not on the validation set. The best obtained accuracy in the article is 92% and 86.7% on the train and validation set, respectively, using their fine-tuned ResNet-50 model. The authors extended this research by developing a continuous recognition model (*Montefalcon, Padilla & Rodriguez, 2023*). Features of 15 Filipino phrases were extracted using MediaPipe and classification is performed using long-short term memory and residual network. Initial result shows that the long-short term memory outperformed the ResNet-34 model, achieving 94% accuracy. Facial feature importance analysis is also explored by the authors. They've shown that excluding features from eyebrows, mouth, and eyes significantly reduced the performance of the model. This provides evidence that non-manual signals are important features for recognition. It is noted, however, that despite excellent performance of these deep networks, computational resources are not always available and an efficient, cost-effective approach is needed (*Wang et al., 2023*). As a solution to this, lightweight

conversion methods are explored by *Cayme et al. (2024)* for their convolutional and long-short term memory networks for FSL recognition. Memory utilization and model size has been significantly reduced while having minimal reduction in performance metrics.

Deep networks are now advanced but still suffer from major drawbacks such as interpretability, transparency, and need for large amounts of data. In this article, we explore another promising field, specifically TDA, for FSL recognition. So far, no work has been developed that used TDA for this problem. Moreover, no article has been observed to have used the dataset in this article, giving this study novelty in the field.

# PERSISTENT HOMOLOGY CLASSIFICATION ALGORITHM (*DE LARA, 2023*)

## Persistent homology

Persistent homology (PH) is a method widely used in topological data analysis (TDA). It can be used for determining invariant features or topological properties of a space of points that persist across multiple resolutions (*Carlsson, 2009*; *Edelsbrunner & Harer, 2008*). These invariant features capture the qualitative properties of data due to their sensitivity to small changes in the input parameters, making PH favored by researchers. PH application extends to different data types such as point clouds, images, time series, *etc*. In this article, we focus on the computation of PH on point clouds.

A point cloud $(X, d)$ represents a finite set of points $X$ together with a distance function $d$. The usual assumption is that $X$ is sampled from an underlying topological space $\mathbb{S}$. However, describing the topology of $\mathbb{S}$ based on sample points of $X$ is not easy. This is where PH can be used.

In computing PH, a filtration is constructed using $X$, converting the point cloud into a nested sequence of simplicial complexes. A visual description of this process is presented in Fig. 1. This is done by defining a non-negative real number $\varepsilon$ that serves as a parameter to thicken $X$. We denote the thickened point cloud corresponding to the parameter $\varepsilon$ as $X_\varepsilon$. As the value of $\varepsilon$ increases, simplices are added to the complexes, and a sequence of nested simplicial complexes is formed.

Adding new simplices to the complexes can be done using a variety of ways. In this article, Vietoris-Rips (VR) complex is used. In VR complex, two points $x_i$ and $x_j$, each of which are initially 0-simplex, are connected when the distance $d(x_i, x_j) \leq 2\varepsilon$. This forms a 1-simplex or a line segment. Adding another point $x_k$ that satisfies $d(x_i, x_k) \leq 2\varepsilon$ and $d(x_j, x_k) \leq 2\varepsilon$ forms a 2-simplex or a triangle. Adding another point $x_l$ that satisfies $d(x_a, x_l) \leq 2\varepsilon$ for $a = i, j, k$ forms a 3-simplex or a tetrahedron, and so on. It is worth noting that the parameter $\varepsilon$ is the only parameter changing in this filtration process. The addition of simplices depends on this $\varepsilon$ value and the distances of 0-simplices from one another.

In each filtration step, the *homology groups* are extracted. These are invariant features of a topological space that provide important information and can be computed algebraically. The homology of the underlying topological space $\mathbb{S}$ can be approximated by the homology of the simplicial complexes derived from $X$.

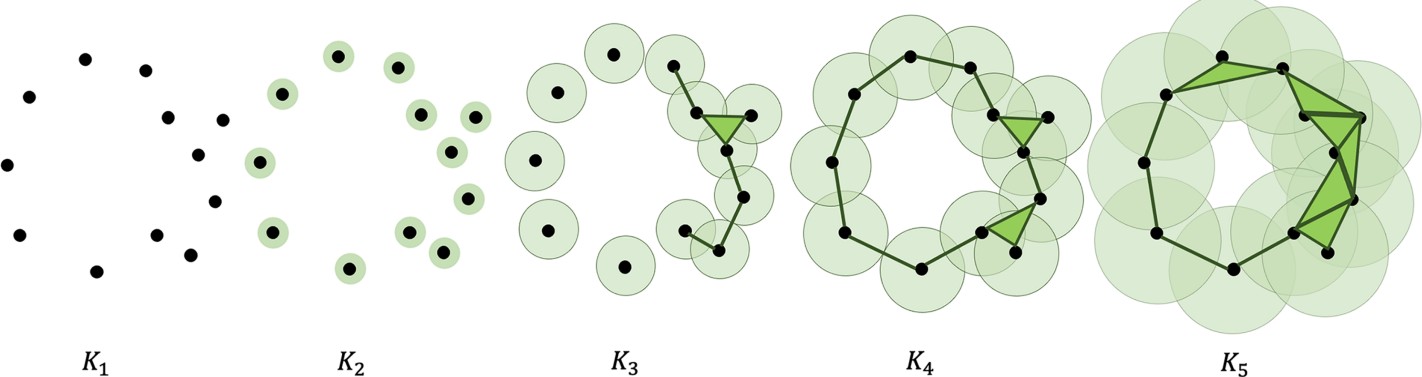

$K_1$ $\qquad$ $K_2$ $\qquad$ $K_3$ $\qquad$ $K_4$ $\qquad$ $K_5$

**Figure 1 Illustration of a filtration of a point cloud into a nested sequence of simplicial complexes with $K_1 \subseteq K_2 \subseteq \ldots \subseteq K_5$.**

For this, suppose $K$ is a finite simplicial complex and $K_1 \subseteq K_2 \subseteq \ldots \subseteq K_r = K$ is a finite sequence of nested subcomplexes of $K$. Here, $K$ is called a filtered simplicial complex and the sequence $\{K_1, K_2, \ldots\}$ is the filtration of $K$. The homology of each of the subcomplex can be computed as follows. For each $k$, the inclusion maps $K_i \rightarrow K_j$ induce $\mathbb{F}_2$-linear maps $f_i^j : H_k(K_i) \rightarrow H_k(K_j)$ for all $i, j \in 1, 2, \ldots, r$ with $i \leq j$. It follows from functoriality that $f_l^j \circ f_i^l = f_i^j$ for all $i \leq l \leq j$.

Now suppose $K_s$ is a subcomplex in the filtration or a filtered complex at time $s$. We define the $k$-th cycle group of $K_s$ as $Z_k^s = Ker\partial_k^s$ and the boundary group of $K_s$ as $B_k^s = Im\partial_{k+1}^s$. Then, the $k$-th homology group of $K_s$ is given by

$$H_k^s = \frac{Z_k^s}{B_k^s} = \frac{Ker\partial_k^s}{Im\partial_{k+1}^s}. \tag{1}$$

Consequently, for $p \in \{0, 1, 2, \ldots\}$, the $p$-th persistent $k$-th homology group of $K$ given a subcomplex $K_s$ is

$$H_k^{s,p}(K, K_s) = H_k^{s,p}(K) = \frac{Z_k^s}{B_k^{s+p} \cap Z_k^s} = \frac{Ker\partial_k^s}{Im\partial_{k+1}^{s+p} \cap Ker\partial_k^s} \tag{2}$$

and the $p$-th persistent $k$-th Betti number $\beta_k^{s,p}$ of $K_s$ is the rank of $H_k^{s,p}$.

Simply, for each nonnegative integer $k$, there exists a $k$-th homology group $H_k(X_\varepsilon)$ representing $X_\varepsilon$. The 0-th dimensional, 1-dimensional, and 2-dimensional homology groups gives the connected components, holes or tunnels, and voids, respectively. These algebraic structures are homotopy invariant, meaning they do not change when the space undergoes bending, stretching, or other deformations, making them ideal as representation of data.

## Persistence diagram and barcode
The result of obtaining the homology of the filtered complexes can be represented using a persistence diagram or persistence barcode. Example of some filtration process and their corresponding diagrams and barcodes are shown in Fig. 2. These representations show the

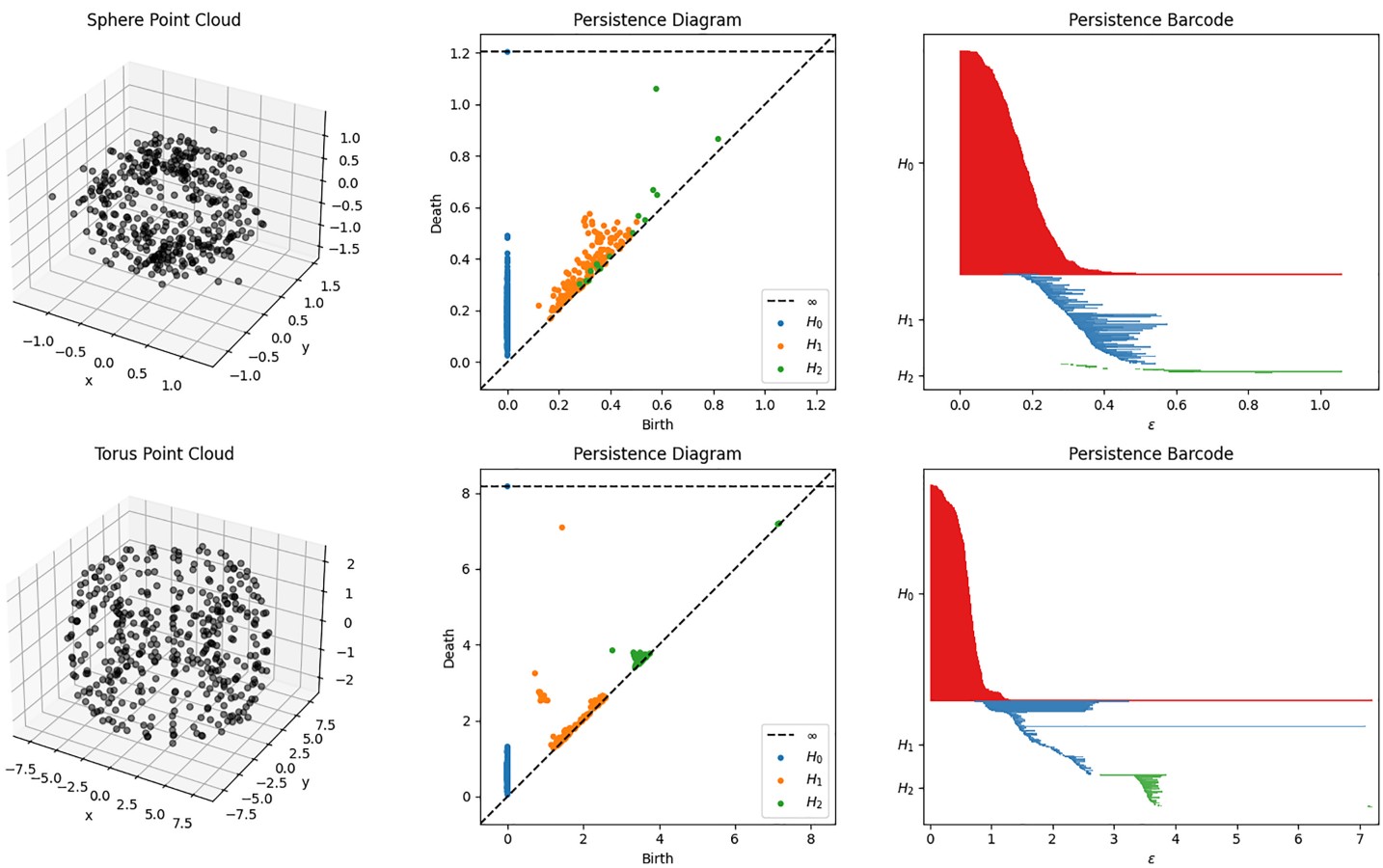

**Figure 2 Sphere and Torus point clouds and their corresponding persistent diagrams and barcodes obtained using persistent homology with Vietoris Rips filtration.** A persistent 2-dimensional hole (void) can be observed from the diagram and barcode of the sphere point cloud. Meanwhile, there is a persistent 1-dimensional hole (tunnel) that can be observed for the torus point cloud.

appearance (birth) and disappearance (death) of intrinsic topological features, such as homology groups and Betti numbers. In other words, these birth and death values represent the filtration index (or the parameter $\varepsilon$) at which the topological feature appear or disappear, respectively. The lifespan or duration of these topological properties are essential for the qualitative analysis of the topology of the data. Shorter lifespan are often associated with noise while longer ones are the important topological features. This lifespan parameter will be essential for the development of the topology-based classifier PHCA. For a more comprehensive discussion of the computation of PH, the reader is referred to *Edelsbrunner & Harer (2008)*.

Alternatively, the persistence diagram and barcode obtained from the point cloud $X$ can also be represented as an $n \times 3$ matrix, denoted as the persistence $\mathscr{P}(X)$. The number of rows $n$ represents the number of topological features or the total number of 0-dimensional holes, 1-dimensional holes, 2-dimensional holes, and so on, depending on the defined

maximum dimensions that can be detected during the filtration. Meanwhile, the first, second, and third column entries of the $q$-th row of $\mathscr{P}(X)$ represents the dimension, birth, and death times of the $q$-th topological feature in the filtration of $X$, respectively.

Now, since that the filtration process cannot be performed in an infinite duration of time, a maximum scale *maxsc* must be defined. Scale in this context represents the $\varepsilon$ value or the distance threshold. In practice,

$$maxsc = \frac{1}{2}\max_{x,y}\{d(x,y)\} \tag{3}$$

is used where $d(x,y)$ is the distance of any two points $x, y$ with $x \neq y$ of the point cloud.

## Training and classification using PHCA

Suppose $X$ is the training dataset consisting of $m$-dimensional data points categorized into $k$ distinct classes. More specifically, suppose that $X = X_1 \cup X_2 \cup \ldots \cup X_k$ where each $X_i$ is the set of data points in class $i$ for $i = 1, 2, \ldots, k$. We note that $X_i \cap X_j$ for $i \neq j$ implying that no two classes contains the same data point. Introduced with a new data point $\alpha$, we want to determine which class does this point belong to.

The training process of PHCA involves computing for the persistence of each class, $\mathscr{P}(X_i)$ for $i = 1, 2, \ldots, k$. Then, the model measures the topological effect of introducing $\alpha$ to each of these classes. For this, the model defines $Y_i = X_i \cup \{\alpha\}$ and compute for $\mathscr{P}(Y_i)$ for $i = 1, 2, \ldots, k$. This process records the changes in PH between $X_i$ and $Y_i$ for $i = 1, 2, \ldots, k$. The new data point is classified to the class which results in the minimum change in PH. This change is measured using the score function discussed in the next section.

## Score function for PHCA

After training the PHCA model, scoring each of the classes is necessary to choose which class does the new data point $\alpha$ belongs. For this, the model computes for $Score(X_i)$ for $i = 1, 2, \ldots, k$ and compare their results. Recall that the PH of a point cloud can be represented as an $n \times 3$ matrix where $n$ represents the number of topological features and the three columns represent the dimension, birth, and death of each topological feature, respectively. From here, we define the lifespan of the $q$-th topological feature as $l_q = d_q - b_q$ where $b_q$ and $d_q$ are the birth and death times of the $q$-th topological feature, respectively. Then, we can define the score function as

$$Score(X_i) = \left| \sum_{q \in \mathscr{P}(Y_i)} l_q - \sum_{q \in \mathscr{P}(X_i)} l_q \right| \tag{4}$$

or the absolute difference of the total sum of lifespan of $\mathscr{P}(Y_i)$ and the total sum of lifespan of $\mathscr{P}(X_i)$. The new data point $\alpha$ is then classified into the class which satisfies

$$\arg\min_i\{Score(X_i)\} \tag{5}$$

## METHODS

In this section, we elaborate on the methodology used in this article which includes the description of the dataset and the overall classification scheme.

### Data description

The dataset used in this article is published by *Porton (2023)* in Kaggle. It consists of 11,700 images of hands that corresponds to FSL signs of a letter in the alphabet. This article only focuses on the static signs. Hence, we omit the images from the letters J and Z.

Each of the 10,800 images considered in this article has dimension 300 pixels × 300 pixels × 3 channels (RGB). There are 450 images for each class or letter of the alphabet, except for J and Z. The images are obtained using a camera that captures a video feed. For each class, the signer performs a sign and a video is captured. Then, frames from the captured feed are extracted as images which forms this dataset. Various lighting and background conditions can be observed in the images, increasing variability in classification.

### Classification scheme

We discuss here the classification scheme implemented in this article. Figure 3 shows the summary of this classification scheme. The main stages are Feature Extraction, Data Splitting, Feature Scaling, Hyperparameter Tuning, and Classification. This scheme is implemented for each trial, totaling up to 10 trials in this article.

#### Feature extraction

The feature extraction process mainly utilized the MediaPipe Hands pipeline (*Zhang et al., 2020*) illustrated in Fig. 3A. The structure of the pipeline involves a palm detection model and a hand landmarker model. The palm detection model reduces the complexity of the task by first estimating a bounding box on the palm. Precise keypoint localization is then implemented using the hand landmarker model to extract the 21 three-dimensional hand-knuckle coordinates. Each landmark consists of $x$ and $y$ coordinates and the $z$-value represents the depth with respect to the camera. Figure 4 presents all hand landmarks detected by the pipeline. For each image in the dataset, 63 keypoints are extracted which will serve as raw features of the image.

Upon extracting the keypoints using MediaPipe, it is observed that not all images are converted into landmarks. Figure 5 summarizes the number of images per class that are converted and not converted by the pipeline. From this analysis, we divide the dataset into two kinds: Balanced and Imbalanced.

For the Balanced dataset, only 313 images are obtained from each of the classes which will be used for classification. Of the classes having more than 313 converted images, the 313 images used in the Balanced dataset are randomly selected. For the Imbalanced dataset, we perform classification on all images converted into landmarks. The number of images per class is observed in Fig. 5.

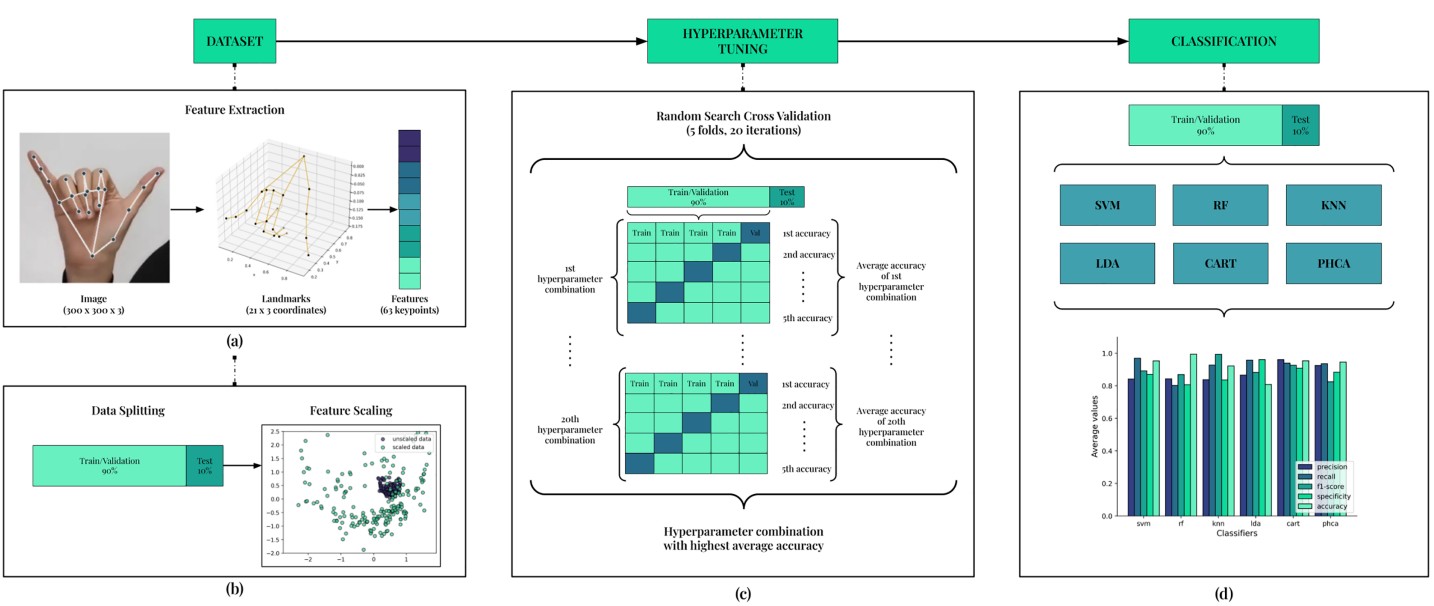

**Figure 3 Framework of the classification scheme.** In (A), 63 features are extracted for each image in the dataset using the MediaPipe Hands pipeline. Then, data is split with stratification into 90–10 train/validation-test sets as shown in (B). Features are then scaled using Standard Scaler. In (C), hyperparameter tuning is employed to ensure best performance for each model. Using the hyperparameter combination that resulted with the best accuracy, the models are trained on the train/validation set and evaluated on the test set using five performance metrics. Comparison is implemented from this result.

0 Wrist
1 Thumb CMC
2 Thumb MCP
3 Thumb IP
4 Thumb TIP
5 Index Finger MCP
6 Index Finger PIP
7 Index Finger DIP
8 Index Finger TIP
9 Middle Finger MCP
10 Middle Finger PIP

11 Middle Finger DIP
12 Middle Filger TIP
13 Ring Finger MCP
14 Ring Finger PIP
15 Ring Finger DIP
16 Ring Finger TIP
17 Pinky MCP
18 Pinky PIP
19 Pinky DIP
20 Pinky TIP

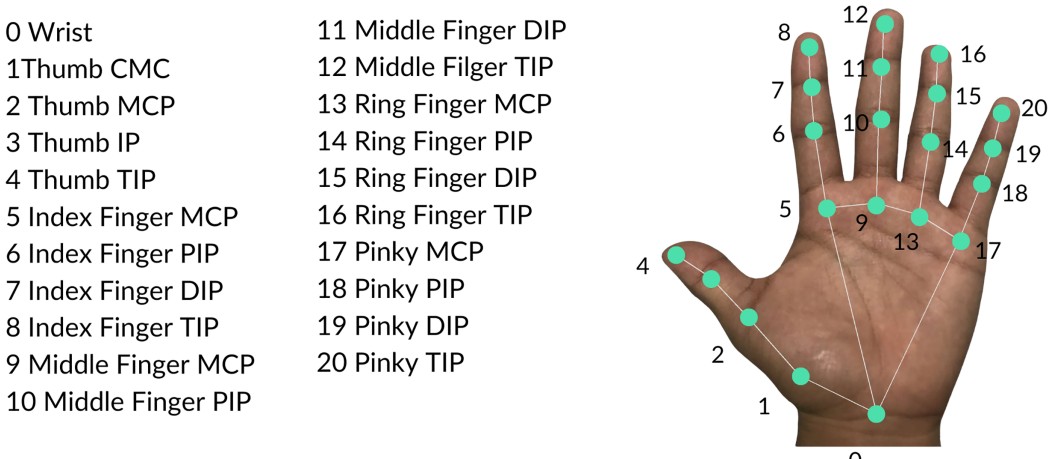

**Figure 4 Hand landmarks detected by the MediaPipe Hands pipeline.**

### Data splitting and feature scaling

Upon extraction, the Balanced and Imbalanced dataset are respectively split into 90–10 train/validation-test set with stratification. This ensures that the proportion of instances for each class is maintained for the train/validation set and the test set. This is crucial especially for the Imbalanced dataset which has different number of instances per class.

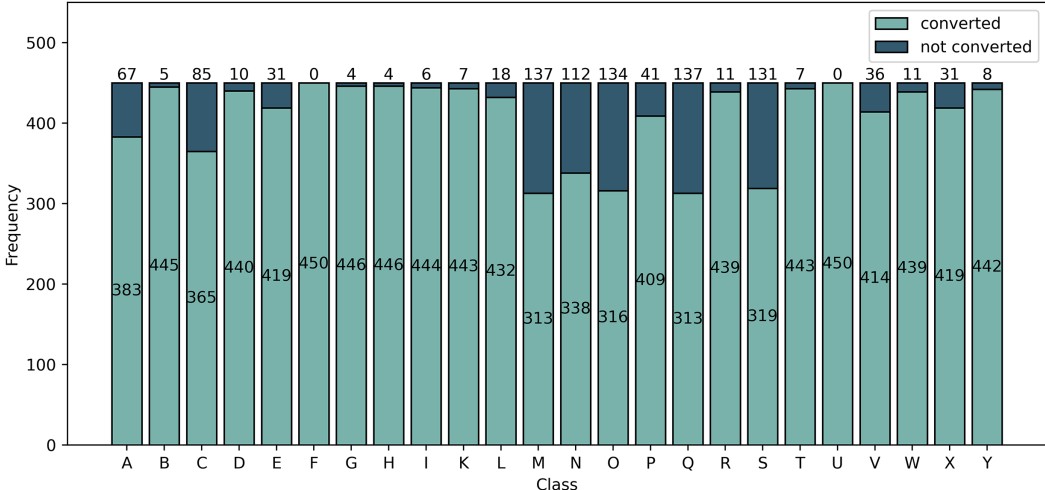

**Figure 5 Frequency of images converted and not converted into landmarks by the MediaPipe pipeline.** Class F and U are all converted into landmarks while class M and Q had the least number of converted images.

From the split data, Standard Scaler is used to scale the data to unit variance and transform the mean of the dataset to 0. The scaler is fit on the training data and same transformation is applied to the test data. This step improves classification by ensuring that the instances in the dataset are similar in scale.

### Hyperparameter tuning

To guarantee an unbiased comparison of models, we implement hyperparameter tuning to each of the models used (see Fig. 3C). In this stage, we utilized Random Search with Stratified K-fold Cross Validation. We implement 20 iterations for the search and employ a five-fold setup for the cross validation.

For each iteration, a combination of hyperparameters is generated randomly for the model from a set of tuning parameters shown in Table 1. Then, the train/validation set is split into five folds. For each cross validation, one of the folds is used for validation and the others are for training. The model with the randomly generated hyperparameter combination is trained using the training folds and evaluated using the validation fold. This results with five accuracy values. The values are then averaged which will serve as the "score" of the hyperparameter combination. In total, there are 20 iterations and correspondingly, 20 average accuracies. The highest average accuracy is obtained and the corresponding hyperparameter combination is used for the model in the classification stage. Ten (10) trials is performed in this article, hence, 10 hyperparameter combinations are obtained.

### Classification

Equipped with the best hyperparameter combination for a given trial, the models are now trained using the 90% train/validation set and evaluated on the 10% test set (see Fig. 3D). The results of this evaluation are five performance metrics values: precision, recall,

**Table 1 Tuning parameters used for each model in implementing hyperparameter tuning. Some of these tuning parameters are lifted from *Nanda & Dutta (2023)*, *Mantovani et al. (2015)*, *Probst, Boulesteix & Bischl (2019)*.**

| Classifiers | Tuning Parameters |
|---|---|
| SVM | C: uniform $(2^{-3}, 2^{15})$ |
| | gamma: auto or scale |
| | kernel: rbf |
| RF | n_estimators: 1–350 |
| | max_depth: 1–5 |
| | min_samples_split: 1–10 |
| KNN | n_neighbors: 1–500 |
| LDA | solver: svd or lsqr or eigen |
| | shrinkage: uniform (0, 1) |
| CART | max_depth: 1–30 |
| | min_samples_leaf: 1–60 |
| | min_samples_split: 1–60 |
| PHCA | homology_dimension: $H_0$ only, $H_1$ only, $H_0$ and $H_1$ |

F1-score, specificity, and accuracy. Further discussion on how these values are computed is given in the following section. These values serve as main basis for comparison of the models, including PHCA.

### Performance evaluation and comparison

To evaluate the performance of the classifiers, five evaluation metrics are obtained, the values of which depend on the confusion matrix corresponding to the predicted classes.

a. **Confusion matrix**

The confusion matrix is a square matrix $A = [a_{ij}]$ where each element $a_{ij}$ represents the number of instances belonging to class $i$ and predicted to be in class $j$. From this confusion matrix, we can obtain the following values: True Positive (TP), True Negative (TN), False Positive (FP), and False Negative (FN).

Aside from these values, the confusion matrix itself can be used for misclassification analysis of a model. In this article, we investigate further the confusion matrix obtained by PHCA to analyze how the model performed on our dataset.

b. **Classification performance**

From the TN, TP, FP, and FN values, we obtain five performance metrics which will comprise the classification performance for each of the models. These are precision, recall, F1-score, specificity, and accuracy. The first 4 is averaged across all classes while the latter is obtained over the entire test set. The description of these metrics are provided in the following:

*i.* **Precision** describes exactness.

$$precision = \frac{TP}{TP + FP} \tag{6}$$

*ii*. **Recall** describes completeness.

$$recall = \frac{TP}{TP + FN}. \tag{7}$$

*iii*. **F1-score** describes the combination of precision and recall, providing insights on the balance of the two metrics.

$$f1score = \frac{2 \times precision \times recall}{precision + recall}. \tag{8}$$

*iv*. **Specificity** describes the ability of the classifier to predict instances not belonging to a class.

$$specificity = \frac{TN}{TN + FP}. \tag{9}$$

*v*. **Accuracy** describes the ratio between the number of correct predictions to the total number of predictions made.

$$accuracy = \frac{TP + TN}{TP + TN + FP + FN}. \tag{10}$$

c. **Comparison of classification performance**

To compare the performance of PHCA with the performance of the other classifiers in terms of the five evaluation metrics, Nemenyi test is implemented. It serves as a *post-hoc* test for the implementation of Friedman test, a non-parametric equivalent of the repeated-measures ANOVA (*Demšar, 2006*). The null hypothesis for the Friedman test states that all classifiers are equivalent. If this is rejected, then pairwise comparison of the classifiers is done using Nemenyi test. The performance of two classifiers is significantly different if the corresponding average ranks differ by at least the critical difference

$$CD = q_\alpha \sqrt{\frac{k(k+1)}{6N}} \tag{11}$$

where $k$ is the number of classifiers, $N$ is the number of datasets, and $q_\alpha$ are based on the Studentized range of statistic divided by $\sqrt{2}$. The threshold value $\alpha$ used in this article is 0.05.

# RESULTS AND DISCUSSION

PHCA and five other classical classifiers (SVM, RF, KNN, LDA, CART) are evaluated on an FSL alphabet image dataset (Balanced and Imbalanced) using five metrics: precision, recall, F1-score, specificity, and accuracy. Ten (10) trials are implemented and for each trial, the hyperparameters of the classical classifiers are tuned using random search with 20 iterations together with stratified five-fold cross validation. In all trials, PHCA with

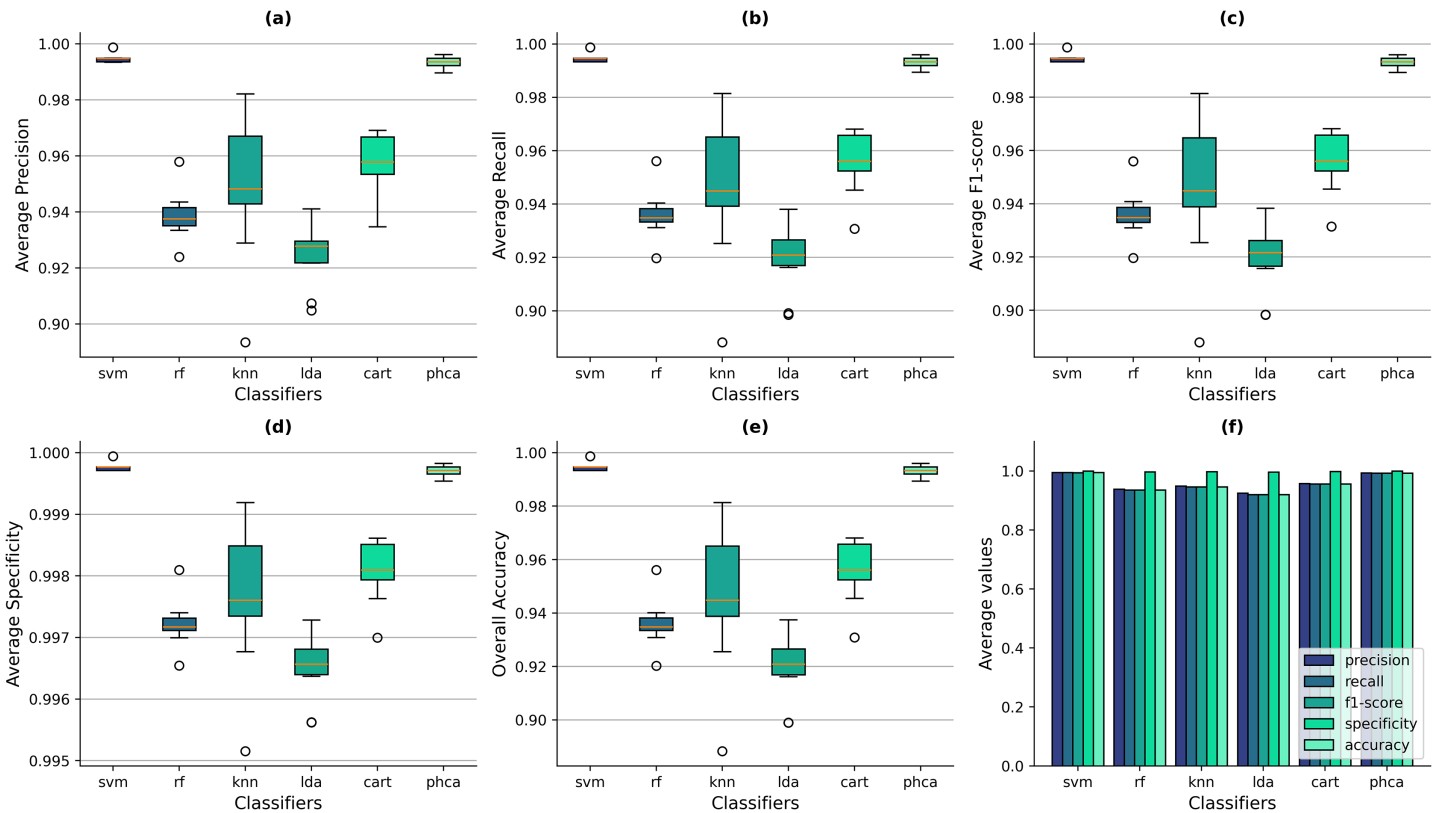

**Figure 6 Distribution of performance metrics obtained by PHCA and the classical classifiers for the Balanced dataset.** Ten (10) trials are implemented, each with a different train-test split. Shown are the box-and-whisker plots of the resulting average precision (A), recall (B), F1-score (C), specificity (D), and overall accuracy (E). Shown also are the average metric values obtained by the classifiers across all 10 trials (F).

homology dimension 0 obtains the best validation accuracy and hence is used for classifying the test set. We analyze the results below that show the potential of PHCA as an alternative for FSL recognition.

## Performance evaluation

Figures 6A–6E show the box-and-whisker plots of the distribution of performance metrics obtained by the classifiers for the Balanced dataset. It can be observed that PHCA and SVM performed better than the other classifiers across all metrics since the interquartile ranges (IQRs) of the two classifiers lie significantly above than others. From Fig. 6E, it shows that the overall accuracy values of SVM ranges from 99.34% to 99.87%, with a mean of 99.45% across all trials. Meanwhile, PHCA's overall accuracy ranges from 98.94% to 99.60%, with a mean of 99.31%. On the other hand, the maximum accuracy values of RF, KNN, LDA, and CART across the trials are 95.62%, 98.14%, 93.75%, and 96.81%, respectively. The same observations can be said for the other metrics as shown in Fig. 6A for the precision, Fig. 6B for recall, and Fig. 6C for F1-score. For the average specificity, the scale of difference in values among classifiers is smaller. SVM's average specificity ranges from 99.97% to 99.99% with a mean of 99.97% while that of PHCA ranges from 99.95% to

99.98% with a mean of 99.97%. Compared to the average specificity values of RF, KNN, LDA, and CART which are 99.72%, 99.77%, 99.65%, and 99.81%, respectively. This implies that RF, KNN, LDA, and CART also performed well in terms of specificity. However, it can be noted that the distribution for specificity values are similar with the other metrics as observed from the box-and-whisker plots in Fig. 6D, which still indicates the superiority of PHCA and SVM. Figure 6F shows the average metric values across the trials of all classifiers. It provides additional evidence that SVM and PHCA performed better than the other classifiers, hence, supporting our claim.

Now, we can also deduce from Figs. 6A–6E that the performance, in terms of all metrics, of SVM and PHCA are not significantly different on the Balanced dataset. This is because for each metric, the median line (50% quartile) of SVM falls within the IQR of PHCA. For the precision values, the median of SVM is 99.48% and the IQR of PHCA is 99.22–99.49%. For the Recall values, the median of SVM is 99.46% and the IQR of PHCA is 99.19–99.47%. Similarly for the F1-score values, the median of SVM is 99.46% and the IQR of PHCA is 99.19–99.46%. For the specificity and accuracy values, the median of SVM is exactly the upper bound of the IQR of PHCA which are 99.97–99.98% and 99.20–99.47%, respectively. We investigate this claim further in the following section.

The IQRs of PHCA in all metrics are small relative to RF, KNN, LDA, and CART but is bigger than that of SVM. This implies that the dispersion of metric values is low, indicating consistent performance across the trials. Comparing this with KNN which has the largest IQR in each metric, implying higher dispersion in metric values. This suggests that the performance of KNN is highly dependent on the dataset and consequently, on its "n_neighbors" hyperparameter. The low dispersion of PHCA provides evidence to the stability of the classifier under changes in the dataset.

Similarly, Figs. 7A–7E show the box-and-whisker plots of the distribution of performance metrics obtained by the classifiers for the Imbalanced dataset. The superiority of PHCA and SVM to the other four classifiers in all setups can still be observed from the figure, based on the location of their IQRs. Figure 7E shows that the overall accuracy values of SVM ranges from 99.39% to 99.80%, with a mean of 99.60% across all trials. Meanwhile, PHCA's overall accuracy ranges from 98.87% to 99.80%, with a mean of 99.40%. To compare, the maximum accuracy values of RF, KNN, LDA, and CART across the trials are 93.76%, 99.28%, 94.27%, and 97.24%, respectively. The scale of difference in the specificity values for the imbalanced dataset is also smaller. SVM's average specificity ranges from 99.97% to 99.99% with a mean of 99.98% while that of PHCA ranges from 99.95% to 99.99% with a mean of 99.97%. Compared to the average specificity values of RF, KNN, LDA, and CART which are 99.73%, 99.97%, 99.75%, and 99.88%, respectively. Despite the excellent performance of these four classifiers in terms of specificity, the box-and-whisker plots in Fig. 7D still shows that PHCA and SVM performed better. The bar plots in Fig. 7F shows also that overall, PHCA and SVM performed better than the other classifiers.

Now, notice in Figs. 7A–7E that the median of SVM in all metrics did not fall within the IQRs of PHCA. This suggests that there is a significant difference between the two classifiers in all metrics. We further investigate this claim in the following section by using some statistical test.

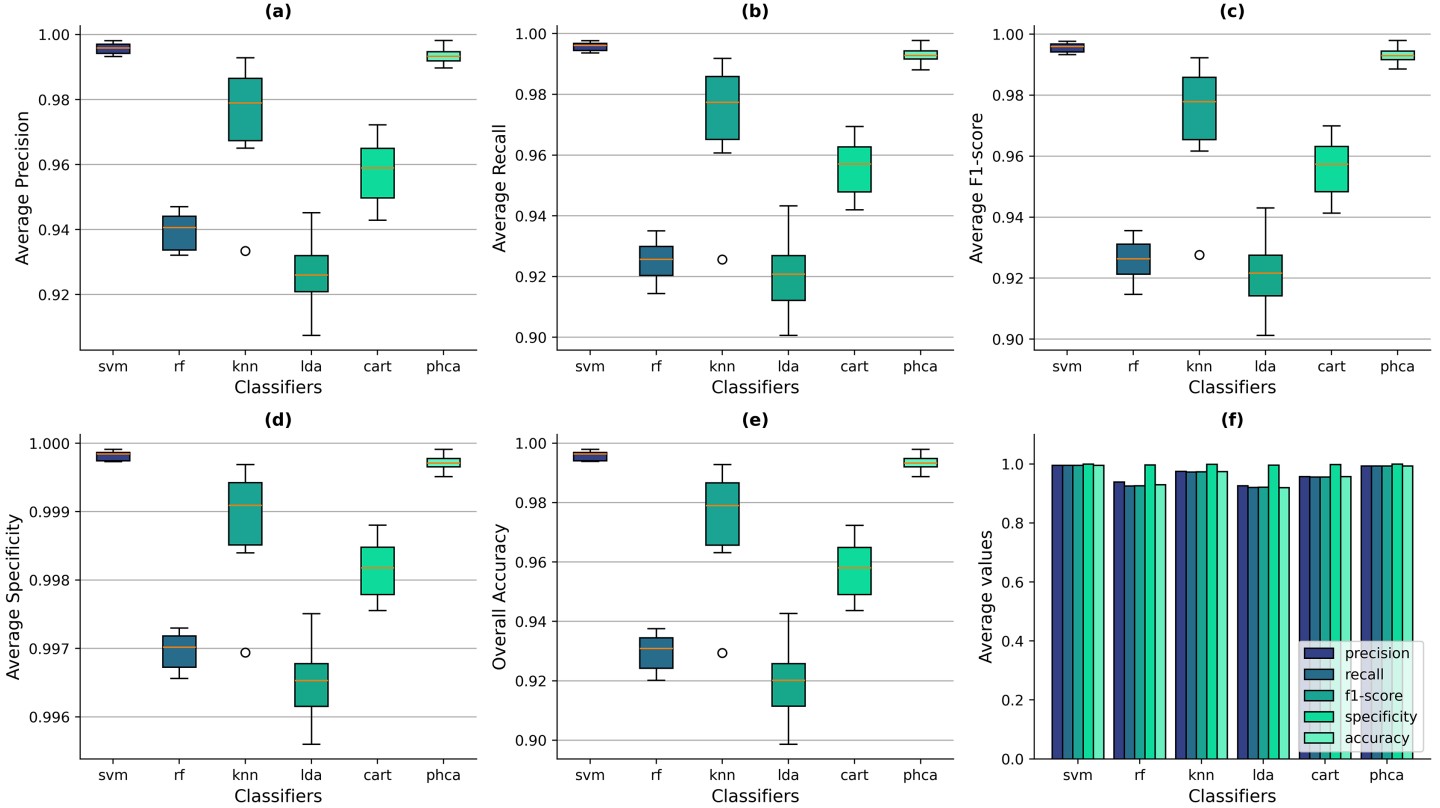

**Figure 7 Distribution of performance metrics obtained by PHCA and the classical classifiers for the Imbalanced dataset.** Ten (10) trials are implemented, each with a different train-test split. Shown are the box-and-whisker plots of the resulting average precision (A), recall (B), F1-score (C), specificity (D), and overall accuracy (E). Shown also are the average metric values obtained by the classifiers across all 10 trials (F).

Compared to the Balanced dataset, the IQRs of each classifier in all metrics also increased for the Imbalanced dataset. This means that the metric values of the classifiers are more dispersed, which is reasonable considering the nature of the dataset which makes classification more challenging. However, the IQRs of PHCA in all metrics are still smaller relative to RF, KNN, LDA, and CART and slightly bigger than SVM. This means that the dispersion of the metric values of PHCA is relatively low which indicates consistency in performance across trials.

## Comparison of classification performance

Now, we empirically compare the classifiers in terms of overall performance, *i.e.*, in terms of all metrics. For this, we state the null hypothesis that there is no significant difference in overall performance and hypothesize that there is. We use Friedman Test to test our claim.

The *p*-value obtained from the Friedman Test is less than the threshold value $\alpha = 0.05$ in all trials for both the Balanced and Imbalanced datasets. Hence, we implement a *post-hoc* analysis using Nemenyi Test to pairwise compare the classifiers. The results are shown in Tables 2 and 3 for the Balanced and Imbalanced datasets, respectively. We focus on comparing PHCA with the other five classical classifiers.

**Table 2 Pairwise comparison result of the Nemenyi test on PHCA against SVM, RF, KNN, LDA, and CART for the Balanced dataset.** If the value is greater than the threshold value $\alpha = 0.05$, then there is a significant difference in terms of overall performance between the compared classifiers. Otherwise, there is no significant difference in overall performance. The bold values indicate the significant performance differences.

| Trial | PHCA-SVM | PHCA-RF | PHCA-KNN | PHCA-LDA | PHCA-CART |
|-------|----------|---------|----------|----------|-----------|
| 1 | **0.9000** | 0.0010 | 0.0010 | 0.0010 | 0.0010 |
| 2 | **0.9000** | 0.0010 | 0.0010 | 0.0010 | 0.0010 |
| 3 | **0.9000** | 0.0010 | 0.0010 | 0.0010 | 0.0010 |
| 4 | **0.9000** | 0.0010 | 0.0010 | 0.0010 | 0.0010 |
| 5 | **0.9000** | 0.0010 | 0.0193 | 0.0010 | 0.0010 |
| 6 | **0.8994** | 0.0010 | **0.0604** | 0.0010 | 0.0010 |
| 7 | **0.9000** | 0.0010 | 0.0010 | 0.0010 | 0.0010 |
| 8 | **0.9000** | 0.0010 | 0.0010 | 0.0010 | 0.0010 |
| 9 | **0.9000** | 0.0010 | 0.0022 | 0.0010 | 0.0010 |
| 10 | **0.9000** | 0.0010 | 0.0010 | 0.0010 | 0.0010 |

**Table 3 Pairwise comparison result of the Nemenyi test on PHCA against SVM, RF, KNN, LDA, and CART for the Imbalanced dataset.** If the value is greater than the threshold value $\alpha = 0.05$, then there is a significant difference in terms of overall performance between the compared classifiers. Otherwise, there is no significant difference in overall performance. The bold values indicate the significant performance differences.

| Trial | PHCA-SVM | PHCA-RF | PHCA-KNN | PHCA-LDA | PHCA-CART |
|-------|----------|---------|----------|----------|-----------|
| 1 | **0.9000** | 0.0010 | **0.4162** | 0.0010 | 0.0010 |
| 2 | **0.9000** | 0.0010 | 0.0010 | 0.0010 | 0.0010 |
| 3 | **0.9000** | 0.0010 | 0.0293 | 0.0010 | 0.0010 |
| 4 | **0.9000** | 0.0010 | 0.0010 | 0.0010 | 0.0010 |
| 5 | **0.9000** | 0.0010 | 0.0193 | 0.0010 | 0.0010 |
| 6 | **0.9000** | 0.0010 | 0.0010 | 0.0010 | 0.0010 |
| 7 | **0.9000** | 0.0010 | 0.0010 | 0.0010 | 0.0010 |
| 8 | **0.5559** | 0.0010 | **0.9000** | 0.0010 | 0.0010 |
| 9 | **0.9000** | 0.0010 | **0.6335** | 0.0010 | 0.0010 |
| 10 | **0.8329** | 0.0010 | **0.4527** | 0.0010 | 0.0010 |

From the tables, the results show that there is no significant difference in terms of overall performance between PHCA and SVM, but the performance of PHCA differs significantly from the other classifiers, except for KNN on some trials. This is true for both the Balanced and Imbalanced datasets. Our claim for the Balanced dataset in the previous section is supported by this result but not for the Imbalanced dataset. However, we remark that the claim in the previous section came from an educated guess based on the box-and-whisker plots. Our results from Nemenyi Test provide empirical evidence against this guess.

In some trials, KNN performed well and has shown similar performance with PHCA and SVM. But we note that the dispersion of metric values of KNN is high and this

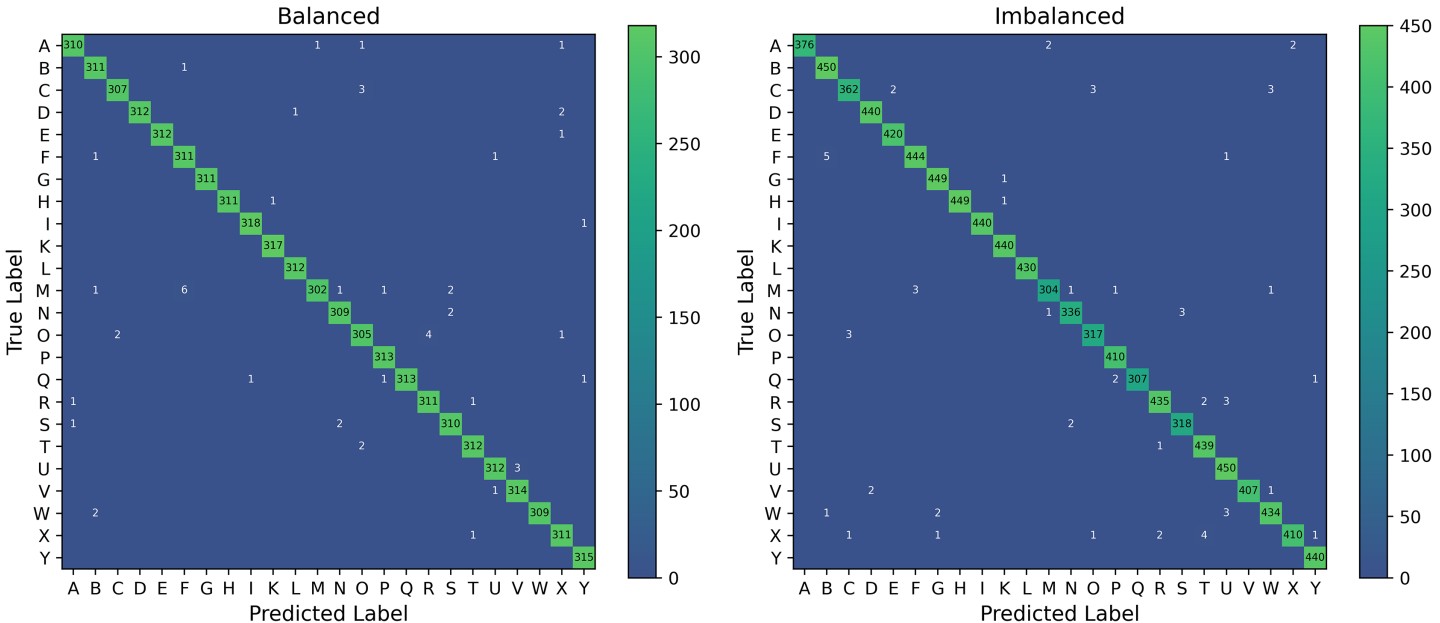

**Figure 8 Confusion matrices associated with the predictions of PHCA for the Balanced and Imbalanced datasets.** The given matrices are the element-wise sum of the confusion matrix for each trial.

similarity of performance is not consistent across all trials. Hence, from the results in Figs. 6 and 7 and the result of comparison in Tables 2 and 3, we can conclude that PHCA and SVM performed best in all setups for both the Balanced and Imbalanced datasets and their is no significant difference in terms of their performance. This highlights that PHCA performance is at par with the best performing classifier for FSL recognition.

## Misclassification analysis of PHCA

We further investigate the distribution of misclassified images by PHCA for the Balanced and Imbalanced datasets. Figure 8 shows the cumulative confusion matrix associated with the predictions obtained by PHCA.

PHCA made the most number of misclassifications on class M from the Balanced dataset, having a total of 11. Of these, most are classified into class F (6 images) and S (2). The second most misclassified by PHCA is class O which are incorrectly predicted to be in class C (2) and R (4). For the Imbalanced dataset, PHCA misclassified class X the most. Images from this class are often predicted to be in class T (4) and R (2). PHCA also misclassified a few images under class C which are usually incorrectly predicted to be in class O (3) and W (3). The classifier also mispredicted five images from class F to class B.

Shown in Fig. 9 are the images from the stated classes where PHCA had the most misclassifications. We can observe that the hand shape in class M are very similar to that of class S. Similarity in hand structure can be observed from class O and C; so are for class X and T. We also observe similar hand patterns from images under class B and F. These similarities in hand shape and structure show evidence for the misclassifications on the

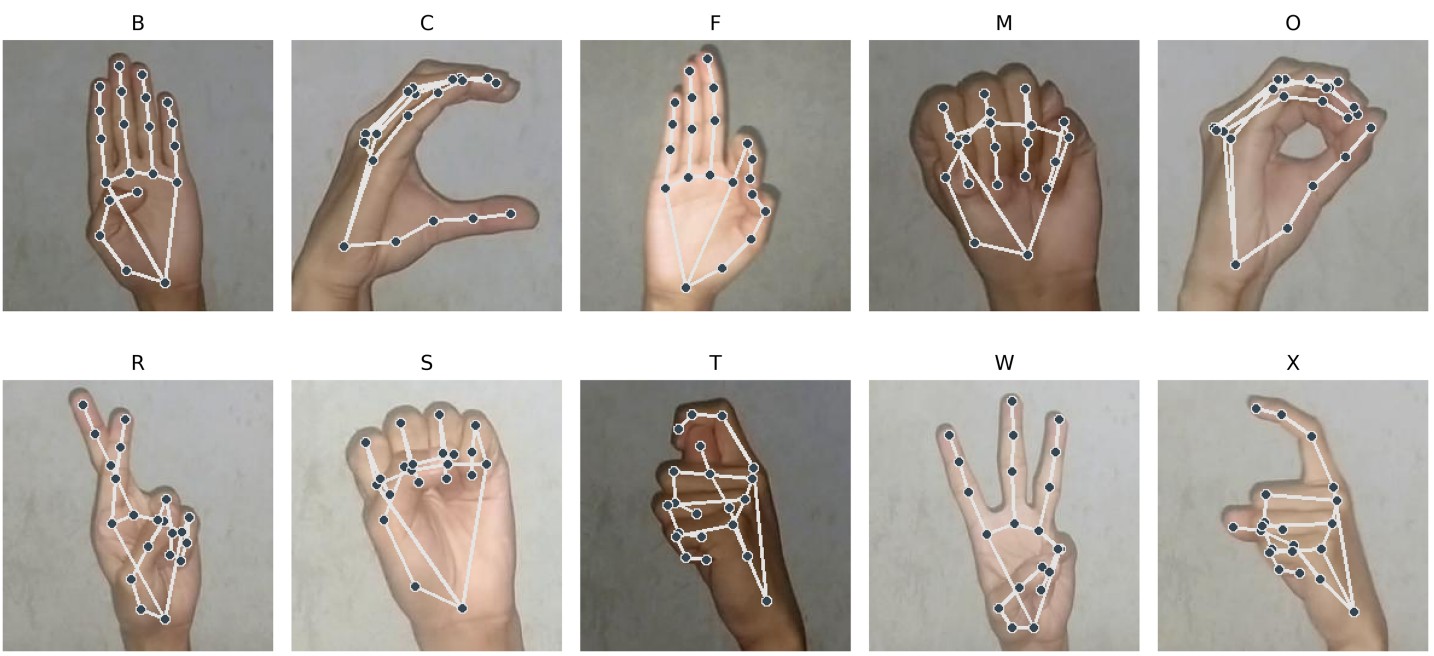

**Figure 9 Sample images with landmarks that are often misclassified by PHCA.**

predictions of PHCA. Similar structures imply similar 3D coordinates corresponding to the landmarks obtained by MediaPipe, hence, resulting in the confusion of the classifier.

However, there are hand structures that are different but are still misclassified, such as for class M and F, class O and R, class X and R, and class C and W. These misclassifications may be due to the nature of the features obtained from the images.

Recall that MediaPipe extracts the 3D coordinates of the landmarks from the hand. However, not all images from the same class are consistent in terms of orientation, structure, and proximity with respect to the camera. This variability may have caused dissimilar hand structures to have similar coordinates. This result is also observed from *Oliva et al. (2018)*. Their analysis shows that by using Cartesian coordinates, misclassification can occur only when the gesture is similar or there is a difference in position.

Additionally, PHCA performed at par, and sometimes better, than SVM in classifying images from particular classes. For classes C, F, H, I, M, R, and T, PHCA and SVM had the same number of misclassifications for the Balanced dataset. The two models also have the same number of misclassified images under class A, H, T, and W for the Imbalanced dataset. Meanwhile, PHCA had less number of misclassifications on class A, O, and X for the Balanced dataset. This provides evidence that on some cases, PHCA can perform better than SVM.

## CONCLUSION

In this study, a novel topology-based classifier called PHCA is utilized to classify images of FSL alphabet. The authors focused on static signs, considering only 24 letters of the alphabet, excluding J and Z. The performance of PHCA and classical classifiers, such as

SVM, RF, KNN, LDA, and CART, are compared using precision, recall, F1-score, specificity, and accuracy. The hyperparemeters of the classical classifiers are tuned to ensure the best performance. It is also obtained that PHCA best performed with 0-dimensional homology groups only as its parameter, providing computational evidence to the procedure in Theorems 2 and 3 of *De Lara (2023)*. The experiment is implemented with 10 trials on balanced and imbalanced datasets for a comprehensive comparison.

Results show that PHCA, together with SVM, performed better than the other classifiers in all trials, both for the Balanced and Imbalanced datasets. SVM achieved a mean Accuracy of 99.45% across 10 trials, while PHCA achieved 99.31%. The interquartile ranges of the box-and-whisker plots corresponding to PHCA and SVM are also small in all trials, implying that the dispersion in metric values of the two classifiers are low which indicates consistency in performance across the trials. This validates the stability of Persistent Homology, the main method used in PHCA, under perturbations in the input data.

Statistical analysis shows that there is no significant difference in the overall performance of PHCA and SVM in all trials. Meanwhile, there is a significant difference in the overall performance of PHCA against the other classifiers. This implies that PHCA performed at par with an excellent performing classifier and better than the others used in this article. This is true for both balanced and imbalanced datasets which shows also the robustness of PHCA against the distribution of instances of a dataset.

PHCA misclassified signs that are similar in hand structure. This is reasonable since similar structure corresponds to similar features. However, the use of Cartesian coordinates of landmarks as features can be improved since such features are not position invariant, requiring additional normalization process before classification. Using Spherical coordinates or computing for angles between hand landmarks as features can be explored which may address this problem.

Overall, PHCA is a potential alternative for FSL recognition. Further exploration of TDA methods can be done for classifying dynamic gestures to include letters J and Z, and words or phrases in FSL. In Human Activity Recognition, a similar field with FSL, Persistence Diagrams are being used to extract topological properties from temporal features of an action (*Yang et al., 2023*). This technique can also be explored for FSL recognition. In this article, although FSL alphabet is used as dataset which does not constitute direct application, results show that TDA-inspired methods such as PHCA can serve as an opportunity for further research, showing insights to its usefulness and practical applications.

### Funding
This work has been supported by the University of the Philippines System-wide Computational Research Laboratory Grant. The funders had no role in study design, data collection and analysis, decision to publish, or preparation of the manuscript.

## Grant Disclosures

The following grant information was disclosed by the authors:
University of the Philippines System-Wide Computational Research Laboratory Grant.

## Competing Interests

The authors declare that they have no competing interests.

## Author Contributions

- Cristian B. Jetomo conceived and designed the experiments, performed the experiments, analyzed the data, performed the computation work, prepared figures and/or tables, authored or reviewed drafts of the article, and approved the final draft.
- Mark Lexter D. De Lara analyzed the data, authored or reviewed drafts of the article, and approved the final draft.

## Data Availability

The FSLAlphabetRecognition-PHCA Repository is available at GitHub and Zenodo:

- https://github.com/ji-chani/FSLAlphabetRecognition-PHCA.

- Jetomo, C. (2025). FSLAlphabetRecognition-PHCA. Zenodo. https://doi.org/10.5281/zenodo.14637416.

The FSL Dataset is available at Kaggle: https://www.kaggle.com/datasets/japorton/fsl-dataset?resource=download.

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
