# Peer review of "Filipino sign language alphabet recognition using Persistent Homology Classification algorithm"

_PeerJ Computer Science, doi:10.7717/peerj-cs.2720_

## Round 0.1 · original submission · Major Revisions

Dear authors,

Thank you for submitting your article. Reviewers have now commented on your article and suggest major revisions. We do encourage you to address the concerns and criticisms of the reviewers with respect to reporting, experimental design and validity of the findings and resubmit your article once you have updated it accordingly.

When submitting the revised version of your article, it will be better to address the following:

1. The abstract does not present the creation or usage of the dataset.

2. Evaluation of the techniques covered in the recent literature is not sufficient. Negative and positive aspects of these techniques should be stated. The lessons learned from the literature is not clearly mentioned. These are the key aspects to motivate for the research and for new researchers who want to tackle the same problem.

3. All reported graphs should be accompanied by some concrete description of the lessons learned from the results reflected in the graph. It is important to explain them in detail and to enrich them with some semantics by showing the reasons for these results, how they can be further improved, etc.

4. Equations should be used with equation numbers in the manuscript. Explanation of the equations should be checked. All variables should be written in italics as in the equations. Their definitions and boundaries should be defined. Necessary references should also be given.

Best wishes,

**Language Note:** The review process has identified that the English language must be improved. PeerJ can provide language editing services - please contact us at [email protected] for pricing (be sure to provide your manuscript number and title). Alternatively, you should make your own arrangements to improve the language quality and provide details in your response letter. – PeerJ Staff

Reviewer 1 ·

Basic reporting

- In Figure 4, the vertical axis of the Persistence Barcode plots do not have a label.
- The bar plots in Figures 5 and 6 would benefit more from having more significant figures. Putting confidence intervals and quartile ranges in the bar plots with different values may be a good visual for additional or alternative analysis. An example can be found in Fig. 11 of Berrar, 2016 (Springer Machine Learning, 2016, V106, pp 941) ["Confidence curves: an alternative to null hypothesis significance testing for the comparison of classifiers"].

Experimental design

- The general details on how the classifier hyperparameters were selected are not discussed and may affect the replicability of the study.

Validity of the findings

- The training details are not provided to show that best effort was given in developing all classifiers. For example, for KNN, what k values, distance function and weighting function were used? For SVM which approach (ovr or ovo), which kernel, and what hyperparameters were used? RF and CART also have tunable hyperparameters. You can also specify the number of components used in an LDA for classification.
- If the classifiers were trained using a fixed set of (hyper)parameters without tuning, how are these parameters representative of a specific classifier's performance?

Additional comments

- The presentation of results and analysis is clear.

·

Basic reporting

- some citations missing in introduction (see my annotations)
- dataset needs more detailed description (see my annotations)
- the section on covering what persistent homology is mathematically needs to be revised for clarity. In particular, it is confusing to use the term "p-th persistent homology" and not use "p" as the dimension. The authors have reused p as a scale index in lines 203-204 but have used it as a dimension in lines 202 and 205. Similarly, the authors have used "k" for a scale parameter in line 203 and then used "k" as a dimension variable in lines 203-204. The definition of the persistent homology group is incorrect right above line 204 (see my annotation). My recommendation here is to use "p" as the dimension of the homology group consistently and use i,j where i,p was being used previously. For a better exposition of persistent homology, see https://www.ncbi.nlm.nih.gov/pmc/articles/PMC6979512/

Experimental design

- my major concern with the experimental design is that it does not address the issue of similar (or even the same hand signs) having radically different persistence diagrams when the two point clouds are unioned together vs viewed separately (see my annotations). This concern does not materialize in the results due to overfitting the dataset and the fact that the test sets and the training sets have similar/same distributions. In short, this concern does not disqualify the manuscript, but it does need to be acknowledged for transparency of the methodology.

Validity of the findings

- I checked the code and the figures and the results seem reasonable
- The substantive change I would like to see is an additional section reporting the distribution of misclassified images by the PHCA method. This would not only serve to enhance the findings of the paper, but also open up avenues for further investigation in utilizing PHCA for hand gesture classification. The requested update is given below from the annotations in the pdf:
“Since the thrust of this paper is not just a comparison of PHCA with other ML methods on the FSL data set, I think it would be a valuable addition to include a deeper investigation into why PHCA performed well and why it didn't. This is especially important in light of my earlier comments about the methodology neglecting the possibility of two of the same hand-signals producing vastly different persistence diagrams when unioned together if one hand-signal was in a different orientation that the other (for example).
Please add a figure and some exposition around which hand signals the PHCA method misclassified the most (in terms of raw numbers and in terms of percentage out of the test sets). For example, it would be helpful to know that when you use PHCA with maxdim=0, you misclassify the "O" hand-signal the most frequently (perhaps because H_1 classes are not considered). You would report that (for example) “O” was misclassified 18 times (on average) (5.3% of the test set “O”s were misclassified). This would be best shown as a pareto diagram over all letters.
I believe this would make a valuable addition to the paper and highlight the strengths and weaknesses of PHCA for sign language classification.”

Additional comments

Overall, I appreciate the author's unique approach to using persistent homology techniques to advancing the area of sign language interpretation. The figures provide helpful guidance to follow the manuscript. The comparison in the results section is carried out well and I appreciate the careful reporting on the relevant statistics to ensure that the differences/similarities in performance were quantified.

Overall, the manuscript needs some improvements in clarity of the mathematics being used and the metrics being defined for PHCA. The results of which letters were misclassified and in what frequency by the PHCA method are a valuable addition that is currently missing.

Most of my comments are annotated directly on the PDF. Please address the concerns in the order of:
1. New figure on PHCA misclassification performance
2. Exposition on what the data set is and how it was generated
3. Clarify the persistent homology mathematics
4. Update citations

Thank you

·

Basic reporting

1. Writing can be improved. I recommend that the authors proofread the whole manuscript to spot grammatical errors.
2. Figure captions should be more descriptive and self-contained.
3. The RRL can be improved. There should be more references on the use of PHCA on other classification applications. Is this the first ML algorithm to utilize the data (since data is published in 2023)?
4. Please make sure to use politically correct terms. For example, some group might find "hearing-impaired" offensive.
5. In the abstract, include some quantifiable results on how PHCA is effective in classifying FSL alphabet. Moreover, mention the impact of this study and why this research is relevant.

Experimental design

1. The premise and objectives are novel and worthy of investigation.
2. The authors should highlight the originality more. Are there other ML algorithms applied to FSL?
3. The classification is quite straightforward. Other methods can easily be applied to the pre-processed data. I suggest that the pre-processing be highlighted more.
4. The authors did not justify how the results of the study is useful in practice. If only the alphabets are classified, how can this be useful?

Validity of the findings

1. The pre-processed data should be made publicly available for reproducibility.
2. The codes used should be made available. I suggest using public repository (such as GitHub) for checking. I am interested to check the code since SVM gave similar results.
3. The description of the data is very lacking. There should be a detailed information on how many of each character was used and description on how they were collected.
4. The limitations of the study should be clearly stated.
5. Future works can also be mentioned.
6. Add a sample implementation of the algorithm. I suggest creating a general algorithm that includes pre processing and classification. Adding a flowchart will also help. Then get a sample image (outside the dataset) and test how your method can classify it.
7. There should be more discussions on the misclassified alphabets. I recommend that the confusion matrix be also shown. This way, the authors can easily check which letters become misclassified with other letters. Insights can be drawn from these.

Additional comments

Work is interesting and can be considered for publication after the comments above are addressed.

---

## Round 0.2 · Minor Revisions

Dear Authors,

It is the considered opinion of two of our reviewers that the submitted paper meets the necessary standards to be accepted. However, we would strongly urge you to revise the paper in light of the minor comments and concerns raised by reviewer 2.

Best wishes,

Reviewer 1 ·

Basic reporting

Based on the revised manuscript and the comments on the reviewer's comments, this reviewer has no further comment on the revised manuscript.

Experimental design

The experimental design has been adequately explained in the revised manuscript.

Validity of the findings

The findings indicate that the prpoposed solution is comparable in terms of performance metrics to SVM.

Additional comments

The proposed methods having comparable performacne to traditional classifiers do have some merits, the main "criticism" is the lack of comparison with the state-or-the-art (SOTA) classifiers like deep learning based models, vision transformers (ViT) and Visual LLMs.

·

Basic reporting

Table 1: is there a type for the hyperparameters of PHCA? (PHCA homology dimension: 0, 1, 0 and 1) Is it supposed to say 0, 1, 0 and 1?

The paper has several wonderful improvements from the last iteration by adding tables and figures.

Experimental design

good

Validity of the findings

Lines 447-448: It is also obtained that PHCA best performed
448 with 0-dimensional homology groups only as its parameter, validating Theorem 2 from (De Lara, 2023). -- I took a look at the theorem from the cited paper and it doesn't make a claim about PHCA having best performance with only 0-dimensional homology. The closest claim that paper makes is in this portion: "particularly the 0-dimensional holes, will come earlier. That is, the early appearance or disappearance of topological features translates to a shorter life span of topological features in the filtration of the complexes. This procedure which is based on Theorems 2 and 3 works particularly well when the point clouds for different classes are disjoint from one another or when a point under consideration is close to a particular point cloud."
Thus, it seems that good performance is attributed to the disjointedness of the target classes, not to the restriction to 0-dim homology groups.

Additional comments

If the authors would like to inspire further research based on their results, my suggestion is to give a taste of why someone would want to pursue PHCA instead of SVM in their pipeline. For example, they conclude with, " In this paper, although FSL alphabet is used as dataset which does not constitute direct application, results show that TDA-inspired methods such as PHCA can serve as an opportunity for further research, showing insights to its usefulness and practical applications." This statement can be made stronger if results such as the computational efficiency of using PHCA were highlighted, if that can be shown. So far, the reader is left with ideas like line 452-453: "SVM achieved a mean Accuracy of 99.45% across 10 trials, while PHCA achieved 99.31%" which begs the question of why pursue PHCA instead of the classical techniques.
To be clear, I think this paper is good for publication, but the above suggestion is written for the authors if they'd like to make their manuscript even stronger.

I noticed on line 465, the authors added a citation "This validates the stability of Persistent Homology, the main method used in PHCA, under perturbations in the input data (Mishra and Motta, 2023)." While the citation is appreciated, it is not the correct place for it since the paper cited refers to the stability of a particular construction of persistent homology using an alternative filtration. The proper place for this citation would be line 70-71, "TDA is favored by researchers since it is easy to scale for larger datasets due to the stability of Persistent Homology under perturbations and noise"

·

Basic reporting

no comment

Experimental design

no comment

Validity of the findings

no comment

Additional comments

The authors have addressed all my comments and suggestions. In my opinion, the article meets the criteria for publication.

---

## Round 0.3 · accepted · Accept

Dear Authors,

Thank you for addressing the reviewers' comments. Your manuscript now seems ready for publication.

Best wishes,

·

Basic reporting

n/a

Experimental design

n/a

Validity of the findings

n/a

Additional comments

all of my concerns have been addressed and the paper is ready for publication.